

# DeepPrecip: A deep neural network for precipitation retrievals

Fraser King[1], George Duffy[2,3], Lisa Milani[4,5], Christopher G. Fletcher[1], Claire Pettersen[6], and Kerstin Ebell[7]

[1]Dept. of Geography & Environmental Management, University of Waterloo, 200 University Ave W, Waterloo, Ontario, Canada
[2]NASA, Jet Propulsion Laboratory, 4800 Oak Grove Dr, Pasadena, 91109, California, USA
[3]Earth and Environmental Sciences, University of Syracuse, 900 South Crouse Ave, Syracuse, New York, USA
[4]NASA, Goddard Space Flight Center, 8800 Greenbelt Rd, Greenbelt, Maryland, USA
[5]Earth System Science Interdisciplinary Center, University of Maryland, 5825 University Research Ct suite 4001, College Park, Maryland, USA
[6]Climate and Space Sciences and Engineering, University of Michigan, Space Research Building, Climate &, 2455 Hayward St, Ann Arbor, Michigan, USA
[7]Institute for Geophysics and Meteorology, University of Cologne, Albertus-Magnus-Platz, Cologne, Germany

**Correspondence:** Fraser King (fdmking@uwaterloo.ca)

**Abstract.** Remotely-sensed precipitation retrievals are critical for advancing our understanding of global energy and hydrologic cycles in remote regions. Radar reflectivity profiles of the lower atmosphere are commonly linked to precipitation through empirical power laws, but these relationships are tightly coupled to particle microphysical assumptions that do not generalize well to different regional climates. Here, we develop a robust, highly generalized precipitation retrieval from a deep convolutional neural network (DeepPrecip) to estimate 20-minute average surface precipitation accumulation using near-surface radar data inputs. DeepPrecip displays high retrieval skill and can accurately model total precipitation accumulation, with a mean square error (MSE) 99% lower, on average, than current methods. DeepPrecip also outperforms a less complex machine learning retrieval algorithm, demonstrating the value of deep learning when applied to precipitation retrievals. Predictor importance analyses suggest that a combination of both near-surface (below 1 km) and higher-altitude (1.5 - 2 km) radar measurements are the primary features contributing to retrieval accuracy. Further, DeepPrecip closely captures total precipitation accumulation magnitudes and variability across nine distinct locations without requiring any explicit descriptions of particle microphysics or geospatial covariates. This research reveals the important role for deep learning in extracting relevant information about precipitation from atmospheric radar retrievals.

## 1 Introduction

Accurate estimates of surface precipitation are highly sought-after as they inform flood forecasting operations, water resource management practices and energy planning Buttle et al. (2016); Gergel et al. (2017). Due to the sparse nature of in situ precipitation measurement networks, remote sensing has become a prominent alternative source of observations for deriving surface precipitation estimates Liu (2008). Ground-based scanning radars are valuable resources as they provide estimates of precipitation over a wider area and at a higher temporal resolution compared to traditional in situ gauges Lemonnier et al. (2019).



Additionally, the size and availability of both vertically pointing and space-borne remote sensing datasets have expanded
greatly in recent decades as a result of technological instrument improvements and new satellite missions (Quirita et al., 2017).

Remotely-sensed radar observations used in empirical, power-law relationships can relate radar reflectivity (RFL) estimates
($Z_e$) to surface snowfall ($S$) or rainfall ($R$) rates (Eq. 1) Matrosov et al. (2008); Kulie and Bennartz (2009); Schoger et al.
(2021).

$$Z = a \times (S/R)^b \tag{1}$$

These radar-based retrievals are powerful tools for filling current observational gaps and have been applied to great effect
in previous literature Levizzani et al. (2011); Hiley et al. (2010). However, these relationships demonstrate an inability to
generalize well to unseen validation data as a consequence of the microphysical particle assumptions (e.g. shape, diameter,
particle size distribution (PSD), terminal fall velocity and mass) used in each relationship's unique derivation Jameson and
Kostinski (2002).

Recent machine learning (ML) approaches have demonstrated improvements in estimating surface precipitation from remotely-
sensed data compared to traditional nowcasting methods Shi et al. (2017); Kim and Bae (2017). Deep learning models have
benefited greatly from the increased observational sample provided by remote sensing missions and have shown skill in learn-
ing complex spatiotemporal characteristics of the underlying datasets Chen et al. (2020b). However, a deep learning surface
precipitation retrieval using vertical column radar data with no spatiotemporal covariates has yet to be developed to our knowl-
edge. Previous ML studies have typically focused on passive microwave and infrared datasets which lack a detailed analysis
of the vertical column structure, or suffer from a limited sample for model training across multiple, distinct regional climates
Adhikari et al. (2020); Ehsani et al. (2021).

In this work, we evaluate the abilities of a novel deep learning precipitation retrieval algorithm trained on vertically pointing
radar (up to 3 km above the surface). The regression model we present (DeepPrecip) is a hybrid deep learning neural network
consisting of a feature extraction convolutional neural network (CNN) front-end and a regression feedforward multilayer per-
ceptron (MLP) back-end. The combination of these two architectures allows DeepPrecip to recognize and learn the nonlinear
relationships between different layers in the vertical column of radar observations and produce an accurate surface precipi-
tation estimate. Through an analysis of feature input combinations, DeepPrecip performance is examined to identify regions
within the vertical column that contain the most important contributions to retrieval accuracy Lundberg and Lee (2017). The
relationships that exist between different layers of the vertical profile (and each atmospheric covariate) can be used to help
inform current and future active radar retrievals of surface precipitation.



## 2 Data

### 2.1 Study Sites

In situ data was collected from 9 study sites (Fig. 1.a) from 2012-2020 Schoger et al. (2021); Kim et al. (2021); Munchak et al. (2022); Skofronick-Jackson et al. (2015); Pettersen et al. (2020); Kulie et al. (2021); Houze et al. (2017); Lahnert et al. (2015); Boudala et al. (2021). Colored markers in Fig. 1.b indicate periods where non-zero surface precipitation was recorded. Study sites were selected based on the required presence of an MRR and collocated Pluvio2 weighted precipitation gauge. Rain, snow and mixed-phase precipitation were recorded, with each site's precipitation phase and intensity distribution of
observations differing based on the regional climate. For instance, Marquette experienced strong lake-effect snowfall while Cold Lake received mostly light, shallow snowfall. Further, due to the above zero temperatures recorded at OLYMPEx, these sites were classified as only experiencing liquid precipitation, while ICE-POP received only solid precipitation.

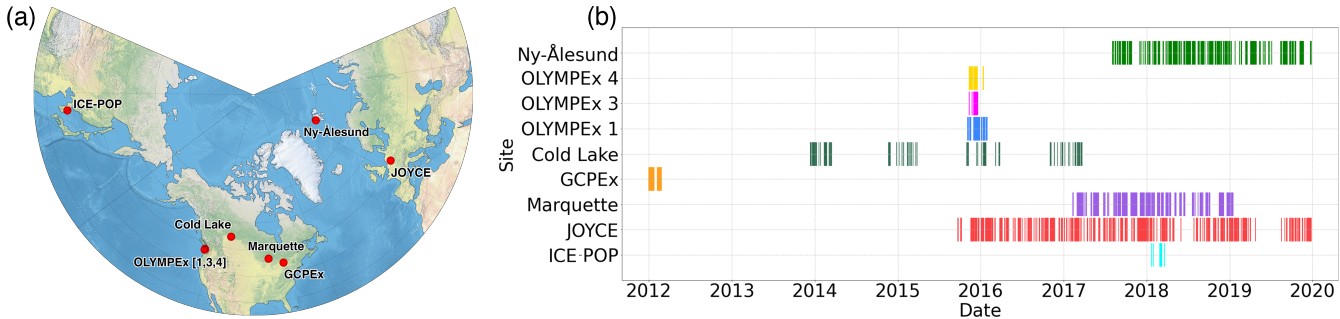

**Figure 1. Observational input data locations and temporal coverage periods. (a),** Geographic study site locations. **(b),** timeline of observational coverage (periods of active precipitation) for each site from 2012 to 2020.

### 2.2 Pluvio2 precipitation weighing gauge

Reference surface precipitation observations were collected by OTT Pluvio2 weighted gauges at each site. The Pluvio2 gauge
records the liquid water content of falling hydrometeors with a time resolution of 1 minute Colli et al. (2014). It includes a $200\ cm^2$ heated surface orifice ($400\ cm^2$ at Ny-Ålesund) to prevent snow and ice buildup, along with site-specific wind shielding implemented as described in Table 1. These fence setups include a Double Fence Intercomparison Reference (DFIR) shield which is a large, double fenced wooden structure which helps significantly reduce the impact of wind on surface precipitation measurements Rasmussen et al. (2012); Kochendorfer et al. (2022). The Alter shield system consists of multiple freely hang-
ing, spaced metal slats around the gauge top opening which also helps mitigate undercatch issues during strong winds Colli et al. (2014). Sensitivity analyses of different rolling temporal windows indicated an optimal temporal resolution of 20-minute non-real time (NRT) accumulation, with minimum observational thresholds of at least 0.2 mm over the course of an hour from the Pluvio2 gauge.



**Table 1.** Summary of in situ study site locations, identifiers, and observational details.

| Site | ID | Lat | Lon | Elev. | Sample ($N$) | Shielding | Source |
|------|----|-----|-----|-------|------------|-----------|--------|
| Ny-Ålesund | 0 | 78.92 | 11.92 | 11 | 19068 | Alter | Schoger et al. (2021) |
| ICE-POP | 1 | 37.67 | 128.7 | 789 | 1705 | DFIR | Kim et al. (2021); Munchak et al. (2022) |
| GCPEx | 2 | 44.23 | -79.78 | 252 | 2314 | DFIR | Skofronick-Jackson et al. (2015) |
| Marquette | 3 | 46.53 | -87.55 | 430 | 8369 | Alter | Pettersen et al. (2020); Kulie et al. (2021) |
| OLYMPEx 4 | 4 | 47.39 | -123.87 | 2155 | 6444 | None | Houze et al. (2017) |
| OLYMPEx 1 | 5 | 47.5 | -123.58 | 3340 | 9114 | None | Houze et al. (2017) |
| OLYMPEx 3 | 6 | 47.68 | -123.38 | 2100 | 5727 | None | Houze et al. (2017) |
| JOYCE | 7 | 50.9 | 6.4 | 95 | 43579 | Alter | Lahnert et al. (2015) |
| Cold Lake | 8 | 54.4 | -110.26 | 541 | 1692 | Alter | Boudala et al. (2021) |

## 2.3 Micro rain radar

Vertical pointing MRRs (developed by METEK) were located nearby the Pluvio2 gauges at each site to record complementary atmospheric observations. The MRR is a K-band (24.23 GHz) continuous wave Doppler radar which provides information related to hydrometeor particle activity up to 3.1 km above the surface (or 1 km for Ny-Ålesund) as a function of spectral power backscatter intensity. The MRR provides 29 vertical bins (of size 100 m) spanning 300 m to 3100 m above the surface as shown for each site in Fig. 2.a. Raw radar measurements were preprocessed using Maahn's improved MRR processing tool

(IMProToo) for noise removal, dealiasing and for extending the minimum detectable dBZ to -14 which allows for improved measurements of solid precipitation. This data was then temporally averaged to align to the same 20-minute windows generated for the Pluvio2 observations Maahn and Kollias (2012).

Ny-Ålesund MRR observations were recorded with a maximal vertical extent of 1 km and are therefore only included in near surface models (additional details in Sect. 4.2). Model skill when including/excluding Ny-Ålesund training data (19,000 sam-

ples) was examined to determine whether it was confounding comparisons between the DeepPrecip subset models presented in Sect. 4.2. We found that the impact on overall performance is negligible across both precipitation phases when Ny-Ålesund is included or excluded in the training set.

## 2.4 ERA5

European Centre for Medium-Range Weather Forecasts Reanalysis version 5 (ERA5) hourly TMP and WVL on pressure levels

from 0 to 3 km were included as additional atmospheric inputs to DeepPrecip Hersbach et al. (2020). These variables were linearly interpolated to align with the 20-minute Pluvio2/MRR datasets with 100 m vertical bin resolutions. ERA5 data was included as a model input to provide additional atmospheric descriptors for DeepPrecip to more accurately recognize different synoptic event structures and precipitation phase states during model training.





**Figure 2. DeepPrecip input covariates, feature processing pipeline and model architecture. (a),** Site-predictor matrix of normalized Micro-Rain Radar (MRR) and ERA5 observational frequency histograms used in model training and testing. **(b),** DeepPrecip convolutional neural network diagram for $L$ inputs with $N$ predictors.




## 2.5 Surface meteorology

Collocated surface temperature (degrees Celsius (° C)) and 10-meter wind speed (m/s) meteorologic observations were also collected from instruments installed at each site and temporally aligned to the Pluvio2 and MRR datasets. Surface wind data acts as an additional observational constraint for mitigating the effects of undercatch on unshielded measurement gauges Rasmussen et al. (2012). We limit the available training dataset to periods when surface wind speeds are $<$ 5 m/s, as this restricts the analysis to low-medium wind speed events at each location.

Surface meteorologic station temperature data is used for precipitation-phase partitioning at $0°$ C to allow for $Z_e - S/R$ comparisons with DeepPrecip. Additional dry surface air temperature thresholds of $1°$, $2°$ and $5°$ C were also examined, but $Z_e - S/R$ performance for both rain and snow appeared optimal when classified using a $0°$ C threshold (where temperatures $< 0°$ C are considered as solid precipitation and temperatures $>= 0°$ C are considered as rainfall). This simple temperature threshold is an additional source of uncertainty in our comparisons with the $Z_e - S/R$ relationships due to the influence of 100 mixed-phase precipitation on power law accuracy, along with uncertainties in the location of the active melting layer Jennings et al. (2018). A more sophisticated phase partitioning system could also be linked to DeepPrecip as an additional predictor to further improve classification of mixed-phase precipitation in future work.

## 3 Methods

### 3.1 Radar-precipitation power laws

Relating radar reflectivity observations to surface accumulation has been done extensively in past surface and spaceborne radar missions through $Z_e - S/R$ power law relationships Skofronick-Jackson et al. (2017); Liu (2008). These power law relationships are empirically defined by relating reflectivity values in a near surface bin to observed surface accumulation under a set of assumed particle microphysics (e.g. size, shape, density and fallspeed) Matrosov et al. (2008). While these techniques have been used to great success in previous studies Schoger et al. (2021); Levizzani et al. (2011), the assumptions 110 about snowfall and rainfall particle microphysics makes the generalization of these power laws less robust, which contributes to high uncertainty when applied across large areas with unique regional climates Jameson and Kostinski (2002).

We examine an ensemble of 12 K-band-derived $Z_e - S/R$ relationships in this work to compare with model output from DeepPrecip (Table 2). As a consequence of the short temporal period (20 minutes) used in this analysis, MSE values are typically small ($< 0.1\ mm^2$). Each $Z_e - S/R$ relationship was applied to a near-surface bin in the reflectivity profile (bin 5 for 115 $DP_{full}$ and $DP_{near}$, and bin 11 for $DP_{far}$) to derive a corresponding surface precipitation estimate. These bins were selected based on a sensitivity analysis where we examined the performance of multiple near-surface high-importance regions of the vertical column (not shown). The best performing regions were identified as the above bins (5 and 11) based on the respective region of the vertical column being considered (near or far). More information regarding the derivation of each $Z_e - S/R$ relationship can be found in Table 2.





**Table 2.** Details for each multi-phase K-band precipitation power law relationship.

| Phase | Name | Power Law | Source |
|-------|------|-----------|--------|
| Solid | AVE_K | $Z_e = 77.61 \times S^{1.22}$ | Schoger et al. (2021) |
| | KB09sp | $Z_e = 19.66 \times S^{1.47}$ | Kulie and Bennartz (2009) |
| | KB09ag | $Z_e = 313.29 \times S^{1.85}$ | Kulie and Bennartz (2009) |
| | KB09br | $Z_e = 24.04 \times S^{1.51}$ | Kulie and Bennartz (2009) |
| | M07 | $Z_e = 56.00 \times S^{1.20}$ | Matrosov (2007) |
| | S17 | $Z_e = 18.00 \times S^{1.10}$ | Souverijns et al. (2017) |
| Liquid | BP09h | $Z_e = 32.00 \times R^{3.30}$ | Van Baelen et al. (2009) |
| | BP09m | $Z_e = 324.00 \times R^{2.40}$ | Van Baelen et al. (2009) |
| | MP48 | $Z_e = 200.00 \times R^{1.60}$ | Marshall and Palmer (1948) |
| | J19bb | $Z_e = 367.00 \times R^{1.37}$ | Jash et al. (2019) |
| | J19nbb | $Z_e = 211.00 \times R^{1.44}$ | Jash et al. (2019) |
| | J19hr | $Z_e = 168.00 \times R^{1.40}$ | Jash et al. (2019) |

To further evaluate the performance of DeepPrecip, we also include model comparisons to a set of six site-derived $Z_e - P$ (reflectivity precipitation) power law relations. Each $Z_e - P$ relationship is empirically derived from the collocated MRR and Pluvio data at each each observational site examined in this work (excluding Cold Lake and Ny-Ålesund due to the limited available sample and vertical extent of each site, respectively). Each $Z_e - P$ relation is fit via a non-linear least-squares approach for finding optimal $a$ and $b$ coefficients in Eq. 1 using SciPy's $curve\_fit$ optimization algorithm (Virtanen et al., 125    2020). Each $Z_e - P$ relationship was then applied to bin 5 reflectivities at each site (i.e. the same process as is used for $Z_e - S/R$ relationships) and compared with in situ observations to assess their general accuracy.

### 3.2    Neural network architecture

DeepPrecip is a feedforward convolutional neural network that takes as input a vector of 115 atmospheric covariates (Table 3), performs a feature extraction of the vertical column and outputs a single surface precipitation estimate using a fully con-
nected multilayer perceptron. While the structure of this final version of DeepPrecip is complex, the retrieval evolved from a much simpler initial state based on a multiple linear regression (MLR) model. Due to clear nonlinearities between observed reflectivity data and surface precipitation accumulation, the MLR model was unable to capture in situ variability and provided estimates near the mean accumulation value. Similar radar-based precipitation retrieval studies by Chen et al. (2020a) and Choubin et al. (2016) have demonstrated much better performance using an ML-based approach which led to the development
of an RF model, an MLP and finally the CNN.

The 1D convolutional layers perform a feature extraction of the vertical column of inputs to reduce the total number of parameters being fed into DeepPrecip's fully connected dense layers. This 1D-CNN structure can identify relationships within





**Table 3.** Summary of DeepPrecip full vertical column model input covariates.

| Predictor | Abbreviation | Count | Units | Source | Type |
|---|---|---|---|---|---|
| Reflectivity | RFL | 29 | dBZ | MRR | float64 |
| Doppler velocity | DOV | 29 | $m/s$ | MRR | float64 |
| Spectral width | SPW | 29 | $m/s$ | MRR | float64 |
| Temperature | TMP | 12 | K | ERA5 | float64 |
| Wind velocity | WVL | 12 | $P_a/s$ | ERA5 | float64 |
| Profile group | PG | 4 | Indicator | K-mean | Boolean |

the vertical column, save on memory and lower computational training time requirements. To perform a 1D feature extraction, the forward propagation step between the previous convolutional layer $(l-1)$ to the input neurons of the current layer $(l)$ are expressed in Eq. 2 Abdeljaber et al. (2017).

$$x_k^l = f(b_k^l + \sum_{i=1}^{N_{l-1}} Conv1d\ (w_{ik}^{l-1}, s_i^{l-1}))\tag{2}$$

Where $k$ and $l$ refer to the $k^{th}$ neuron for layer $l$ with $x$ as the resulting input and $b$ as the scalar bias. $s$ and $w$ terms represent the neuron output and kernel weight matrix respectively, from the $i^{th}$ neuron of layer $l-1$ (and to the $k^{th}$ neuron of layer $l$ for $w$). The function '$f()$' represents the activation function used to transform the weighted sum into an output to be used in the following network layer.

The RF model tested in this study was based on previous work (not shown) where a RF was used to retrieve surface snow accumulation from a collocated X-band and Pluvio2 instrument at a single experiment site (GCPEx). The RF developed in said study demonstrated good skill in estimating surface accumulation, and so we incorporate the same model here as a baseline comparison to other ML retrieval methods (i.e. DeepPrecip).

The final DeepPrecip model structure is outlined in Fig. 2.b. It includes two 1d-convolutional layers, a 1d max pooling layer, dropout layer, flattening layer and concludes in a dense MLP regressor with 3 hidden layers. The total number of trainable model parameters in DeepPrecip is 3,937,793. Model training and testing was performed using a 90/10 (non-shuffled) split on each site to generate training and testing datasets for each location. The non-shuffled nature of this splitting process allows for DeepPrecip estimates to be validated against unseen data and prevents overfitting from training on temporally autocorrelated vertical column inputs. Additionally, this stratified selection process guarantees that an equal percentage of data is included from each site during training.



## 3.3 Hyperparameter optimization

DeepPrecip was developed, trained and optimized on Graphcore intelligent processing units (IPUs) Louw and McIntosh-Smith
(2021) which significantly sped up the training time by a factor of 6.5 compared to a state-of-the-art NVIDIA Tesla V100
GPU. Additional training throughput comparisons are included in Table 4. Training was completed using a combination of
open-source Python packages including Keras, Tensorflow and scikit-learn. Adam optimization is used to minimize a standard
MSE loss function (Eq. 3) to track model learning over time.

$$L = \sum_{i=1}^{D} (x_i - y_i)^2 \tag{3}$$

**Table 4.** DeepPrecip model training throughput comparisons running on Tensorflow (v2.4.3) using a batch size of 128 samples on different
hardware.

| Hardware | Processors | Samples/second |
| --- | --- | --- |
| Graphcore Intelligent Processing Unit (IPU) | 2 | 500 |
| NVIDIA Tesla V100 Tensor Core GPU | 1 | 77 |
| Google Tensor Processing Unit (TPU) | 1 | 56 |
| NVIDIA Tesla K80 GPU | 1 | 23 |

Hyperparameters do not change value during training (in contrast to model parameters like internal node weights), but they
play a critical role in the neural network learning process to map input features to an output. Selecting optimal hyperparameter
values is an important part in constructing a model which minimizes loss, improves model efficiency and quality, and mitigates
overfitting. Multiple steps were taken to address concerns of model overfitting. In addition to the use of non-shuffled training,
we employ multiple regularization methods including early stopping, dropout, L2 regularization and the application of layer
weight constraints (additional details in Table 5).
To select the optimal values for the aforementioned hyperparameters, and to optimize DeepPrecip's general structure, we
use a form of hyperparameterization known as hyperband optimization Li et al. (2017). Hyperband is a variation of Bayesian
optimization which intelligently samples the parameter space to find hyperparameter values that minimize loss while learning
from previous selections. Hyperband adds an additional component to the analysis by also slowly increasing the number of
epochs run during each phase of the optimization process to sample in a more efficient manner. DeepPrecip hyperparameters
were derived by running a 10-fold CV hyperband optimization continuously on Graphcore IPUs for approximately two weeks.
The final values (and their respective parameter search spaces) can be found in Table 5.



**Table 5.** DeepPrecip hyperparameters optimization details.

| Hyperparameter | Value | Parameter Space |
|---|---|---|
| Activation | ReLU | ['relu', 'tanh', 'sigmoid'] |
| Batch Size | 128 | [64, 128, 256, 512] |
| Dropout Rate | 0.1 | [0.001, 0.01, 0.1, 0.25, 0.5, 0.75] |
| Early Stop Patience | 8 | [4, 8, 16, 32] |
| Epochs | 512 | [64, 128, 256, 512, 1024] |
| Filters | 256 | [4, 16, 64, 128, 256] |
| Hidden Layers | 3 | [1, ..., 20] |
| Kernel Size | 16 | [2, 4, 8, 16, 32] |
| L2 Regularization | 0.5 | [0.001, 0.01, 0.1, 0.5] |
| Learning Rate | 1e-7 | [0.001, 0.0001, 1e-5, 1e-7] |
| Loss Function | MSE | ['MSE'] |
| Neurons | 256 | [64, 128, 256, 512, 1024] |
| Optimizer | Adam | ['Adam'] |
| Pool Size | 2 | [2] |

## 3.4 Unsupervised classification layer

An unsupervised k-means clustering preprocessing step is also applied using MRR reflectivity profiles as input to provide
DeepPrecip with insights into distinct profile group (PG) vertical column structures (Fig. 3). Minimizing within-cluster sum
of squares between each vertical column radar estimate results in $k = 4$ PGs being selected using the within-cluster-sum
of squared errors elbow criterion method. K-means clustering was applied using Python's scikit-learn package on all input
reflectivity data to generate four profile clusters which were included as additional input parameters to DeepPrecip. These
clusters are useful for partitioning the precipitation data into groups based on different precipitation event-types (trace, low,
medium and high intensity) to identify where DeepPrecip finds the most significant contributors to high retrieval accuracy for
each category of storm intensity.

## 4 Results

### 4.1 DeepPrecip retrieval performance

Retrieval accuracy is primarily assessed using a mean squared error (MSE) skill metric calculated between each model's esti-
mated surface accumulation values and the corresponding Pluvio2 reference observations over 20 minutes. We first examine
the differences in performance between DeepPrecip and a random forest (RF) that has demonstrated good performance in





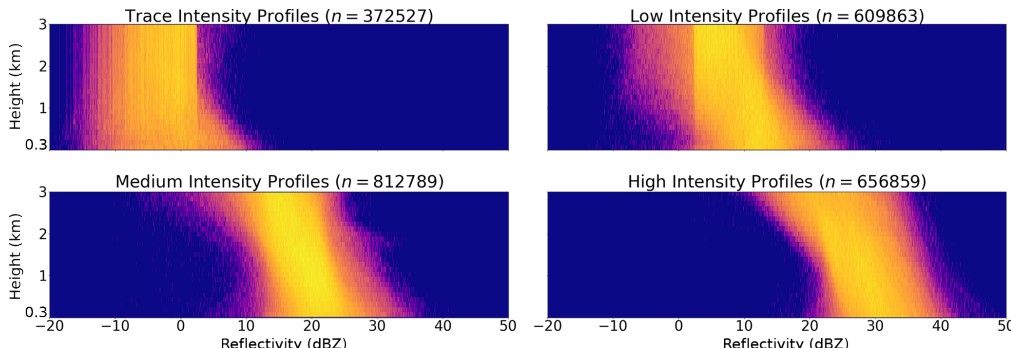

**Figure 3. K-means cluster groups of vertical reflectivity profiles from the MRR instruments at all sites.** A total of 2452038 vertical profiles are organized by reflectivity intensity (dBZ) into $k = 4$ precipitation intensity subsets. The four groups were selected using the within-cluster sum of square elbow method.

our previous work (not shown) to assess the capabilities of a less-sophisticated ML-based approach over a CNN. DeepPrecip demonstrates improved skill in capturing most of the peaks and troughs in observed precipitation variability (Fig. 4.a). Performance statistics (Fig. 4.b) summarize these improvements with DeepPrecip showing MSE values $40\%$ lower and $r^2$ values $40\%$ higher, along with a $0.2$ increase in Person correlation coefficient (significant at $\alpha < 0.05$) compared to the RF.

Total cumulative surface accumulation comparisons between DeepPrecip and each $Z_e - S/R$ relationship are then examined in Fig. 4.c for both rain and snow. To examine model skill across different precipitation phases, a simple temperature threshold is imposed where retrievals recorded during periods with temperatures below zero $^\circ$ C are classified as snow and periods equal to or warmer than zero $^\circ$ C as rain. DeepPrecip more accurately captures surface precipitation quantities when compared to the $Z_e - S/R$ estimates, with a total accumulation curve similar in shape to that of in situ indicating that DeepPrecip more

closely captures the observed precipitation variability and magnitude. Log-scale MSE statistics are calculated between each model and in situ records in Fig. 4.d and indicate that DeepPrecip consistently outperforms traditional power-law methods by $99\%$ on average.

As a general precipitation retrieval algorithm trained on a variety of different storm event-types, we emphasize that we do not explicitly train a DeepPrecip$_{snow}$ and DeepPrecip$_{rain}$ model for different precipitation phases with unique regional atmo-

spheric microphysical conditions. While the $Z_e - S/R$ models shown in Fig. 4.c/d are bespoke for rain or snow, DeepPrecip is trained on all data with no a priori knowledge of the underlying physical precipitating particle state.

DeepPrecip estimates of accumulated rain display a lower MSE than that of snow (Fig. 4.d). We believe these differences to be twofold: 1) the larger sample of rainfall events in the training data (10 times that of snowfall); and 2) the more complex nature of snow particle microphysics. Unlike the uniform properties of a rain droplet, the shape, size and fallspeed of solid

precipitation is much more dynamic and challenging to model Wood et al. (2013). Continued issues with interference from wind may have also impacted the accuracy of in situ measurements of snow accumulation leading to higher uncertainty and error (further discussions on these uncertainties in Sect. 5) Kochendorfer et al. (2017). To visualize the range in uncertainty





from the CNN model estimates, we display 95% confidence intervals in Fig. 4.b/d from 50 DeepPrecip model realizations using dropout. Both ML-based models exhibit the highest uncertainty during periods of mixed-phase precipitation at GCPEx

and Marquette along with high intensity precipitation at OLYMPEx.

To further evaluate DeepPrecip's retrieval skill over traditional methods, we compare model performance to a set of six custom $Z_e - P$ site-derived power laws (derivation details in Sect. 3). While $Z_e - P$ relationships typically perform well in the regional climate under which they were derived, they do not generalize well outside of said climate. This lack of robustness is visible in the accumulation comparisons in Fig. 5.a where the light shaded gray lines (which represent each $Z_e - P$ relationship)

display consistent positive and negative biases. For instance, OLYMPEx 1 and OLYMPEx 3-derived relationships produce a strong positive bias at JOYCE, and the JOYCE-derived $Z_e - P$ power law is quite negatively biased when applied at OLYMPEx. The mean of all six custom power laws is shown in bold gray, and while it closely captures total mean accumulation across all sites, it is unable to model the high variability in precipitation intensity.

The resulting MSE from the application of each custom $Z_e - P$ relationship to each site (along with DeepPrecip) further

demonstrates DeepPrecip's improved robustness (Fig. 5.b). In all cases, DeepPrecip either outperforms all other $Z_e - P$ power laws or is only slightly worse than the power law derived for the site in which it is being tested. Figure 5.b also displays a model intercomparison of each $Z_e - P$ relation, where we can clearly see how $Z_e - P$ relations like those derived at OLYMPEx 1 and 3 are clearly unable to capture the vastly different snowfall regimes at sites like ICE-POP, GCPEx and JOYCE with their much larger MSE values for these sites.

The robustness of DeepPrecip was further evaluated using a leave-one-out cross validation (CV) for each site of training observations. This approach tests the skill of DeepPrecip at predicting precipitation for a location that was not included in the training data, which is a strong indicator of the generalizability of the model. Log-scale MSE results of this test for each site are shown in Fig. 6 for each precipitation-phase subset, along with the corresponding average $Z_e - P/S/R$ estimate when applied at that site. These findings demonstrate similar performance to the baseline DeepPrecip model skill, which continues to

outperform all traditional power law techniques on average. The large range in skill in the power law relationships at most sites (wide error bars) further demonstrates the relative lack of generalizabiltiy of $Z_e - P/S/R$ relationships to different regional climates. Further, the site-derived power law fits (gray dots) perform worse on average than DeepPrecip for locations that are close in proximity (i.e. the OLYMPEx sites).

Predictably, DeepPrecip performance degrades compared to the baseline model when the testing site is left out since the

model is no longer trained using data representing the regional climate of the site being tested. This difference in performance is most notable at the set of OLYMPEx sites, and while DeepPrecip performance is still improved over the $Z_e - S/R$ relationships, we note a substantial percentage increase in MSE (375% on average) at these locations. OLYMPEx measurements were the only observational datasets without any gauge shielding and which is a likely source of uncertainty further contributing to this increase in error when the site is removed from the training set (Kochendorfer et al., 2022).



**Figure 4. Performance comparisons between DeepPrecip (DP), a random forest and an ensemble of power law-derived retrievals of surface precipitation. (a),** Running mean (window size 500 time steps) of accumulation for all sites with Pluvio2 measurements in black, RF estimates in green and DeepPrecip in yellow. Data is sorted by station and then time, with each station separated by a dashed vertical line. Non-rolled values displayed by opaque shaded region and precipitation phase indicated by the colored bottom bar (red is snow, blue is rain). **(b),** performance statistics for RF/DeepPrecip accuracy including MSE, Pearson correlation ($r$) and $r^2$ with error bars showing 95% confidence intervals for $n = 50$ model realizations using dropout. **(c),** Normalized timeseries of total accumulation estimates over the full observation period for all $Z_e - S/R$ relationships and DeepPrecip. The mean of the $Z_e - S/R$ relationships is shown in bold. **(d),** Phase-partitioned log-scale MSE values between each model and in situ observations from 50 model realizations (95% CI in black bars).



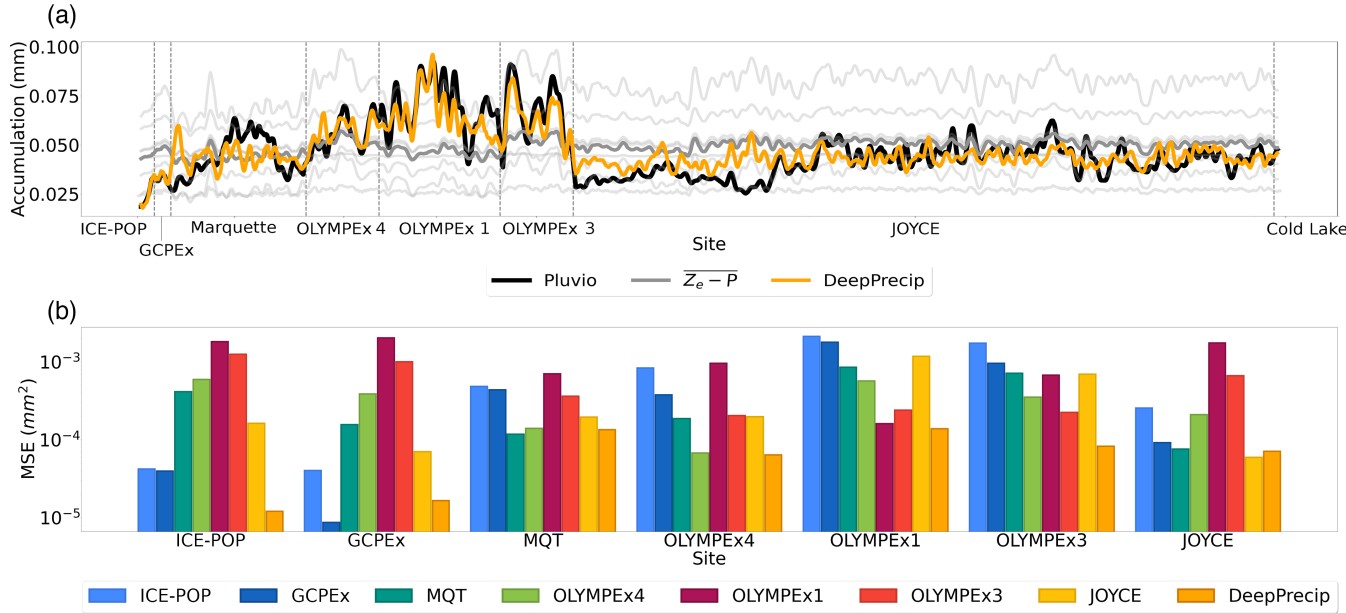

**Figure 5. Site-derived empirical $Z_e - P$ power law performance comparisons. (a),** The same as Fig. 4.a, except now using $Z_e - P$ relationships derived at each study site. **(b),** MSE values for DeepPrecip and each $Z_e - P$ relationship when tested on each site.

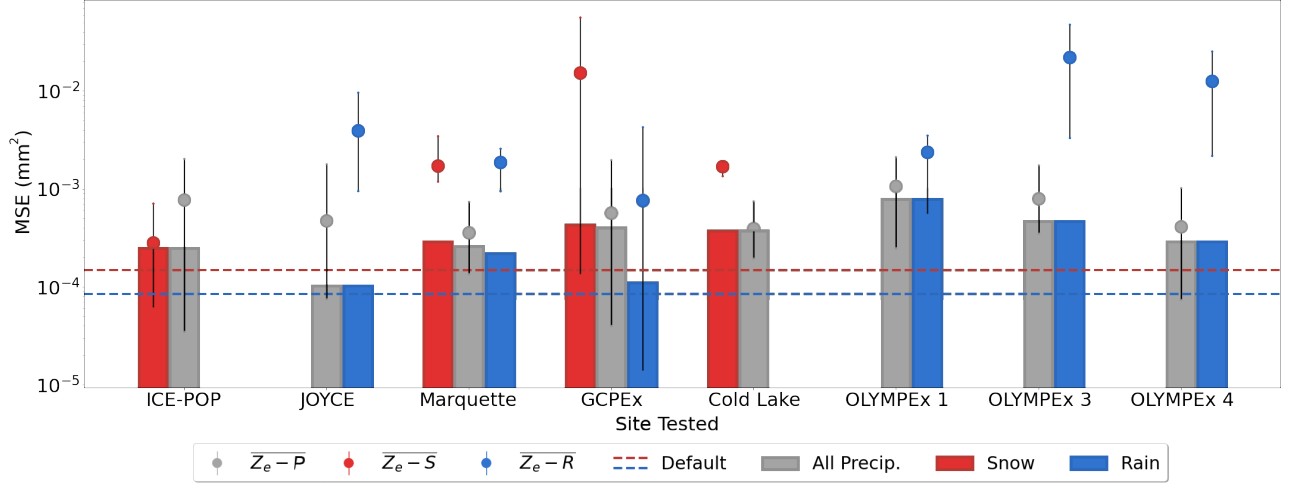

**Figure 6. Leave-site-out full column DeepPrecip performance robustness analysis.** Each bar represents a DeepPrecip full column log-scale MSE value when trained on all precipitation data excluding the noted site, and then validated against said excluded site (dashed line is the default DeepPrecip model with all sites). Each red and blue dot represents the average $Z_e - S/R$ relationship estimate tested in the same manner (error bars represent the min and max ensemble values). Gray dots represent the mean, min and max ensemble values from all site-derived $Z_e - P$ relationships (excluding the relationship derived from site being tested), when applied to each site.





## 4.2   Quantifying sources of retrieval accuracy

Identifying regions within the vertical column that are the most significant contributors towards retrieval accuracy is important for informing future satellite-based radar precipitation retrievals. The ground-based radar instruments used in this work do not suffer from the same ground clutter contamination issues typical of satellite-based radar observations and we are therefore able to quantify the contributions to model skill arising from the included boundary layer reflectivity measurements in DeepPrecip. Separating the training data into three subsets based on vertical extent and generating new models with this data, allows us to examine changes in performance as a function of information availability. These subsets include: $DP_{full}$ (all 29 vertical bins, i.e. the baseline model), $DP_{near}$ (the lowest 1 km; 8 bins), and $DP_{far}$ (1-3 km; 21 bins). DeepPrecip MSE results (Table 6) for each subset suggest that the information provided by a combination of both near-surface and far-profile data results in the highest accuracy.

Distributions of surface precipitation anomalies appear distinct for rain and snow (Fig. 7), with the full column model more closely capturing accumulation recorded by in situ gauges. Anomaly frequencies are derived by removing the mean accumulation estimate for each phase at each site. We attribute the structural differences between the anomaly distributions of of snow and rain to the more complex particle size distributions (PSDs) of snowfall coupled with the more variable particle water content of snow compared to that of rain (Yu et al., 2020). Additional uncertainties in the surface Pluvio2 measurement gauge observational records of snowfall due to gauge undercatch is another likely contributor of increased error (Kochendorfer et al., 2022). In Fig. 7.a, both $DP_{far}$ and $DP_{near}$ exhibit higher anomaly values with a flattened curve top and heavy tails. Using a combination of information from both near and far bins reduce these biases and tightens each accumulation anomaly distribution around zero. A similar trend is also present for rain in 7.b, where we again most closely capture the in situ anomaly distribution using $DP_{full}$.

A major challenge in deep learning is interpreting model output. SHapley Additive exPlanations (SHAP) Lundberg and Lee (2017), is a game theory approach to artificial intelligence model interpretability based on Shapley values that has previously been used to great effect in the Geosciences Maxwell and Shobe (2022); Li et al. (2022). Shapley values quantify the contributions from all permutations of input features on retrieval accuracy to identify which are the most meaningful. While computationally expensive (with exponential time complexity), this process provides local interpretability within the model by examining how each possible combination of all input features impacts model accuracy Jia et al. (2020). Here, the calculated Shapley values give insight into the regions of the vertical column that are contributing the most useful radar information in the precipitation retrieval.

Shapley values for the entire dataset used in $DP_{full}$ indicate that the most important model predictors comprise a combination of both near-surface and far profile bins (Fig. 8). Reanalysis variable model inputs are generally the least influential, except for the trace precipitation case where low-mid level temperature (TMP) and vertical wind velocity (WVL) bins appear highly significant (Fig. 8). In all cases, TMP and WVL decrease in importance as a function of height above the surface. DeepPrecip typically considers MRR-derived bins in the 1.5-2.5 km range as the most significant predictors. In non-trace intensity profiles, it is the 2 km region Doppler velocity (DOV) observations which are the dominant contributing predictor. When we consider





**Table 6.** MSE values (in $e^{-3} \ mm^2$) for all vertical extent experiments across all models for both solid and liquid precipitation.

| Phase | Model | Mean Squared Error ($e^{-3} \ mm^2$) | | |
|---|---|---|---|---|
| | | Full Column | $< 1$ km | $1 - 3$ km |
| Solid | DeepPrecip | 0.051 | 0.07 | 0.063 |
| | RF | 0.073 | 0.11 | 0.075 |
| | AVE_K | 0.74 | 3 | 0.83 |
| | KB09ag | 0.57 | 0.48 | 0.66 |
| | KB09br | 3.7 | 18 | 3.5 |
| | KB09sp | 4.2 | 16 | 4 |
| | M07 | 1 | 6.6 | 1.1 |
| | S17 | 11 | 99 | 10 |
| Liquid | DeepPrecip | 0.044 | 0.055 | 0.077 |
| | RF | 0.095 | 0.053 | 0.1 |
| | BP09h | 7.9 | 8.3 | 5.1 |
| | BP09m | 0.86 | 0.9 | 0.58 |
| | MP48 | 2.2 | 2.6 | 1.1 |
| | J19bb | 1 | 1.2 | 0.96 |
| | J19nbb | 2.2 | 2.8 | 1.2 |
| | J19hr | 3.5 | 4.5 | 1.6 |

all profiles, reflectivity (the input to $Z_e - S/R$ relationships) is not necessarily the dominant feature, and it is a combination of
1.5-2 km profile information from reflectivity, Doppler velocity and spectral width (SPW) that results in the highest model skill.
Combinations of these regions within the vertical column appear to allow DeepPrecip to better understand precipitation events
with complex cloud structures which would not necessarily be recognized by conventional $Z_e - S/R$ relations that primarily
rely on information from a small subset of near-surface bins.

## 5 Discussion and Conclusions

DeepPrecip not only demonstrates considerable retrieval accuracy without the need for physical assumptions about hydrome-
teors or spatio-temporal information, but also provides insight into the regions of the vertical column which are most important
for improving predictive accuracy. The results from Sect. 4.2 suggest that while the exact altitudes providing predictive in-
formation from the vertical column may shift up or down under different precipitation intensities, there exists a consistent
combination of both near-surface and far profile bins that always appear as highly significant contributors to model skill. Fur-
thermore, while RFL is typically considered as the most important predictor in radar-based precipitation retrievals Stephens





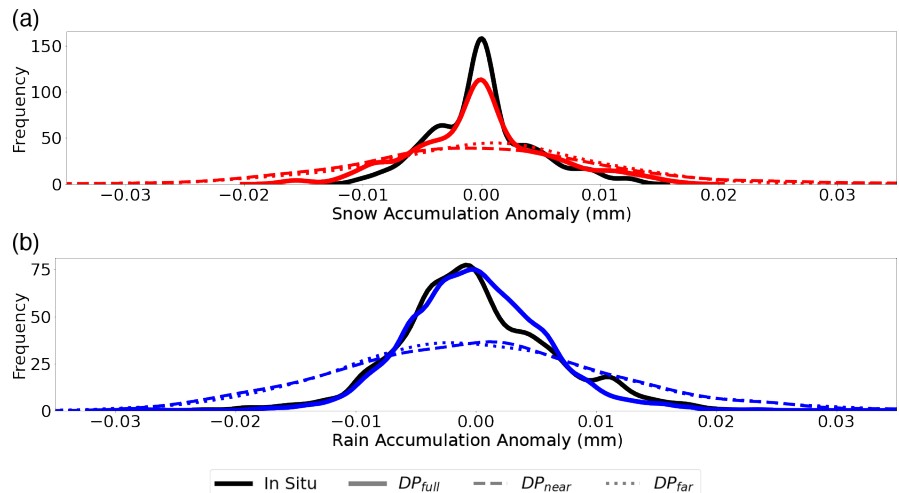

**Figure 7. Phase-partitioned surface precipitation accumulation anomaly frequency distributions.** DeepPrecip is trained and tested on three subsets of bins from the vertical column: $DP_{near}$ ($< 1$ km), $DP_{far}$ ($1 - 3$ km) and $DP_{full}$ (the entire vertical column) for **(a),** solid and **(b),** liquid precipitation.

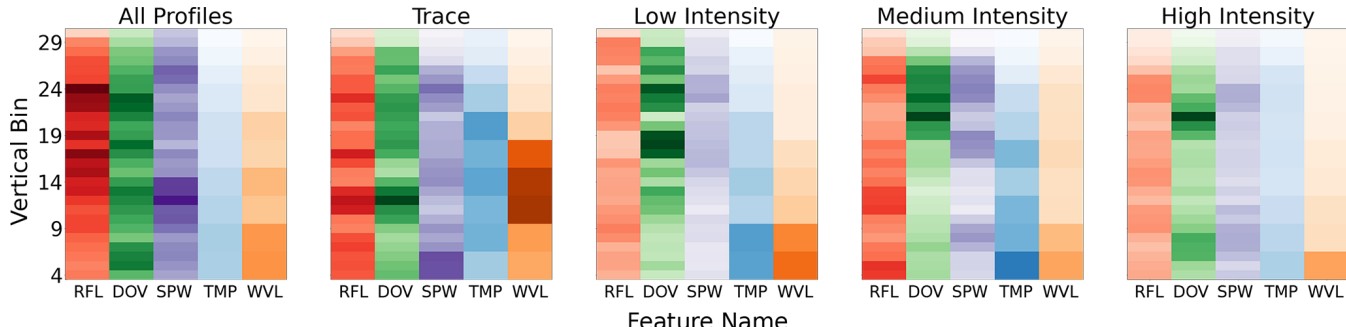

**Figure 8. Normalized vertical column Shapley global feature importance values (i.e. $\overline{|SHAP_{DP}|}$).** Shapley output values are calculated for different subsets of vertical column reflectivities separated into all profiles, trace intensity, low intensity, medium intensity, and high intensity precipitation events based on a k-means clustering of input data (more in Sect. 3.2). Areas of dark color indicate a high feature importance at that location within the vertical column.



et al. (2008); Skofronick-Jackson et al. (2015), we find that contributions from RFL, DOV and SPW provide a near-equal level of importance, with respective average percent contributions to model output of $30\%$, $31\%$ and $28\%$, while ERA5 TMP and WVL variables have a total combined importance of $10\%$.

The combined insights from DeepPrecip's multi-model vertical extent evaluations and feature importance analyses demonstrate a potential to influence current and future remote sensing precipitation retrievals using deep learning. Instruments like CloudSat's Cloud Profiling Radar (CPR), or the Global Precipitation Measurement (GPM) Core's Dual-frequency Precipitation Radar (DPR) also use active radar systems to perform similar, radar-based precipitation retrievals based on data from vertical column reflectivities Stephens et al. (2008). While CPR and GPM-derived products use a more sophisticated Bayesian retrieval to the $Z_e - S/R$ relationships evaluated here, the resulting precipitation estimates are still tightly coupled to a priori physical assumptions of particle shape, size and fallspeed which is a substantial source of uncertainty Hiley et al. (2010); Wood et al. (2013). Additionally, the results of this study further confirm the assumption that there exist regions of high importance in the $< 1$ km (near-surface) region of the vertical column relating to shallow-cumuliform precipitation which influences retrieval accuracy. This is an area that is typically masked in satellite-based products (i.e. the radar "blind-zone") due to surface clutter contamination, and has been shown in previous work to likely be a major source of underestimation from missing shallow cumuliform precipitation Maahn et al. (2014); Bennartz et al. (2019). This work motivates future active radar space-based precipitation algorithms to strongly consider assimilating non-attenuated near surface radar data as additional model inputs to enhance retrieval accuracy, where available.

DeepPrecip is not without uncertainty and error which will reduce its accuracy when tested against new data. Uncertainties present in the training data (stemming from the MRR, ERA5 or Pluvio2 observations), will propagate through the model and bias the output estimates Kochendorfer et al. (2022); Jakubovitz et al. (2019). We have taken steps to mitigate the impact of these uncertainties through multiple data alignment and preprocessing decisions (details in Sect. 3), however precipitation gauge undercatch, wind shielding configurations, MRR attenuation and differences in site-specific vertical extent cannot be eliminated as contributors of retrieval error. Furthermore, while the collection of data from multiple sites provides us with a robust training set under multiple regional climates, due to the unique experimental setups at each site, calibration biases between study locations may further reduce DeepPrecip's skill when applied to new data. As the MRR instrument has a limited 3 km maximum vertical range, we also miss deep convective precipitation events occurring outside of the atmospheric boundary layer, which may contribute to further surface precipitation underestimation. Internal CNN model uncertainty is likely driven, in part, by a combination of the high variability that is typical of precipitation and the limited sample from nine measurement sites over 8 years, which does not fully capture all different forms of possible precipitation structure and occurrence.

*Code and data availability.* DeepPrecip example code is fully open-source and available for download and use on the project's public GitHub repository (https://github.com/frasertheking/DeepPrecip). In situ data is freely accessible for download on Zenodo (https://doi.org/10.5281/zenodo.5976046). ERA5 hourly atmospheric data can be downloaded for free from the Copernicus Climate Change Service (C3S) Climate Data Store. The MRR and Pluvio data used in this study at OLYMPEx, GCPEx and ICE-POP were provided by NASA's Global Precipitation



Measurement (GPM) Ground Validation program. POC: David B. Wolff, David.B.Wolff@nasa.gov. MRR and Pluvio data for the Cold
Lake site was provided from ECCC observation sites. POC: Robert Crawford, Robert.Crawford@ec.gc.ca. JOYCE MRR and Pluvio data
was provided by the University of Cologne. POC: Stefan Kneifel, skneifel@meteo.uni-koeln.de. Ny-Ålesund MRR and Pluvio data were
provided by the German Alfred Wegener Institute for Polar and Marine Research and the French Polar Institute Paul Emile Victor. POC:
Kerstin Ebell, kebell@meteo.uni-koeln.de. Marquette MRR and Pluvio data is provided by the Climate and Space Sciences and Engineering
group at the University of Michigan. POC: Claire Pettersen, pettersc@umich.edu.

*Author contributions.*  Project concept by FK and GD, data provided by FK, CP, KE and LM, methods developed by FK, GD and CF, model
design by FK, data processing and experiments performed by FK, manuscript writing from FK, editing from FK, CF, GD, LM, CP and KE.

*Competing interests.*  The authors declare that they have no conflict of interest.

*Acknowledgements.*  This study was supported by a grant from the Canadian Space Agency Earth System Science: Data Analyses fund.
Further support was also provided by the Natural Sciences and Engineering Research Council of Canada. We also thank the data suppliers:
Environment and Climate Change Canada (ECCC), the National Aeronautics and Space Administration (NASA), the Institute for Geophysics
and Meteorology (IGM) at the University of Cologne, the Korean Meteorological Administration (KMA), the German Alfred Wegener
Institute for Polar and Marine Research (AWI), the French Polar Institute Paul Emile Victor (IPEV), and the Climate and Space Sciences
and Engineering group at the University of Michigan. This research would not have been possible without data contributions from David
Wolff (DW), Claire Pettersen (CP) and Kerstin Ebell (KE). Finally, we would like to thank Graphcore for providing access to their high
performance computing systems for training and optimizing the model.

KE appreciates the funding by the Deutsche Forschungsgemeinschaft (DFG, German Research Foundation) - Project-ID 268020496 -
TRR172. The JOYCE data was provided by the Cloud and Precipitation Exploration Laboratory (CPEX-LAB, http://cpex-lab.de), a compe-
tence centre within the Geoverbund ABC/J.



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
