# Peer review of "DeepPrecip: A deep neural network for precipitation retrievals"

_EGUsphere, 2022_

## Referee Comment (RC1)

**DeepPrecip: A deep neural network for precipitation retrievals.**

King et al. (2022) submitted to AMT

**Review by:**

Anonymous Reviewer

**Introduction and Recommendation:**

Precipitation is a vital measurement for the earth sciences and humanity because of its direct impacts on human life and property though flooding as well as its in direct effects of water resources for human consumption and agriculture. Despite the importance of precipitation measurement, measuring precipitation on a global scale has been a challenging endeavor and has been continuously pursued by NASA for more than two decades now (i.e., TRMM & GPM). The most direct and likely most accurate method of measuring precipitation is through rain gauges, where the amount of liquid water equivalent precipitation can be measured. The drawback of gauges is that they are only point measurements and precipitation has high spatio-temporal variability. Thus, remote sensing efforts can alleviate in-situ measurement drawbacks, but can suffer from their own suite of issues (e.g., calibration, DSD assumptions etc.). Thus there is plenty of room for improvement of these remote sensing techniques.

The authors contribute to the remote sensing literature by presenting a remote sensing method to derive the surface precipitation accumulation. To do the retrieval, the authors they take advantage of a quickly growing method in meteorology/atmospheric sciences, machine learning (c.f., Figure 1 Chase et al. 2022). More Specifically, the authors use vertically pointing radars (i.e., MRR) paired with surface rain gauge measurements from various global locations to train a deep learning retrieval to map measured radar values to the surface precipitation value. In their analysis they show that their new method, DeepPrecip, is able to outperform a random forest method and more simple power-law relationships (which are predominately the legacy method of going from radar measured data to precipitation rate/accumulation).

The paper is generally well written, and the authors are knowledgeable on the methods used in the paper. Furthermore, this paper does fit the scope of AMT and would be a valuable contribution to the literature after addressing the few comments I have made below. I am formally designating these comments as **major** revisions because I am not sure if they will take more than 2 weeks to implement or not (this is the designation between major and minor in my head).

**Major comments:**

**Units of the accumulation data**

The main concern I have is with the main result of the paper and the units of the data in Figure 4. To me these accumulation values seem unreasonably small, which makes me concerned there is some sort of error in either a unit conversion or these are truly just really light precipitation events. Let me explain my reasoning.

[Figure]

*Figure 1: Figure 4 from the paper with my annotation of the maximum accumulation I read from the figure (Black Star and lines).*

The maximum accumulation reported in Figure 4a (and Figure 6a) shows a 20 min accumulation of 0.1 mm. I am more familiar with English units, so 0.1 mm is 0.004 inches of precipitation. 0.004 inches of precipitation is less than the precision of a common ASOS tipping bucket (1 tip in a tipping bucket gauge is 0.01 inches of precipitation). Thus most of the precipitation events shown in Figure 4 are instances where the amount of precipitation in 20 mins is less than 1 tip of an ASOS rain gauge. I hope you see my concern if these accumulations are correct. If the maximum precipitation amount is less than 1 tip of a tipping bucket gauge how representative is the data the authors present compared to the 'global' distribution of precipitation, especially convective precipitation.

While we are discussing Figure 4a, do you have any thoughts of why for JOYCE and Marquette the ML predictions are basically anchored around 0.04 mm?

One last comment on Figure 4, (specifically c). I am confused how this plot was made. What is a normalized time step? Could you help readers by explaining a bit more how this plot was made in the text?

*Transparency of which dataset results are from:*
While I am happy the authors made sure they spent the time to explain their hyperparameter search and some details of their training/test splits, there is no explicit comment of which dataset is being shown in the results section. This is vital to any machine learning paper. The authors must state if the results being shown are from the training or the test dataset. This will allow readers to assess if the results are an unbiased assessment of skill or if they seem to be overfit.

This gets more challenging since the authors did a k-fold cross validation approach, I am unsure which fold they used to show the results. Please explain.

*Table 2 and Literature power-laws:*

Upon inspection of the references provided for Table 2, I noticed that some of these are not specifically K-band relationships. For example Kulie and Bennartz (2009) derive relationships for W-, Ka- and Ku- but not K. Similarly, Matrosov (2007) is for Ka and W. Lastly, Marshall and Palmer (1948) is a Rayleigh power law. While it might seem like using a Ka-band relationship for K-band is harmless, issues arise when non-Rayleigh conditions are encountered (e.g., particle sizes are similar to the wavelength), which tend to coincide with large precipitation rates. You should acknowledge that this could be a source of error in your analysis and might be an unfair comparison for your discussion on lines 195 – 209 (Figure 4cd).

Lastly, you state in the table caption and the discussion on line 112 that all the relationships are K-band, which is incorrect. To prevent future readers from inaccurately using the reported relationships in your paper on their K-band radars, please correct this mistake.

*Random Forest model details:*

The authors mention a random forest model that was based on previous work, but no citation is provided for this model. Given that this random forest model is involved in the primary conclusions of this paper, there is more detail needed. The current description on Lines 146 – 149 is insufficient for reproducibility do not include any of the details of how big the random forest is. Please provide the citation where this model was developed. If there is no citation, please provide more specific details on the random forest model. Also, please note if this model was re-trained on your current data or is it still using the X-band snowfall relationships from GCPEX.

**Minor comments:**

*Scaling of data:*
There was no discussion on if you scaled the features of the ML model. It is common practice to scale data to have mean 0 and variance 1 in machine learning so that the ML model doesn't unintentionally use a variable with a larger absolute value. For example the dynamic range of radar reflectivity is -10 – 40 dBZ. While the range of temperature is 233 – 313, and the range of doppler velocity is -5 – 5 m/s. See how these three all vary on a different order of magnitude?

Did you end up scaling your data? Or did you use batchnorm in training? (I did not see this a parameter in Table 5). Please comment on this.

*Citation issues:*

It would seem there was an issue with LaTex building the document, all of the parenthetical citations do not correctly put citations into parentheses. This made reading some parts of the paper more difficult. Please be sure to use \citep[e.g.,][]{Paper} to correctly get the formatting to work. (or \citet for inline citations).

**Line by Line Comments:**

Please note that word change suggestions are suggestions! Please do not feel pressured to accept my recommendations.

Line 16: This is an example of the citation issue noted in the minor comments.

Lines 60: Be careful here. In my head liquid water content is usually the water content per cubic meter (e.g., g/m^3). Might be good to use a different word here, "records precipitation accumulation" something like that.

Line 84: could you spell out what TMP and WVL are? This is the first time they are defined

Figure 2: I assume darker colors mean higher density? You might want to either include a colorbar somewhere or write it in the caption. Could you note in the caption that the wind velocity is vertical wind velocity and which direction negative is? (is negative wind velocities up or down?). This confused me at first because I thought it might be the horizontal wind velocity, but then I didn't know how to interpret negative values. Why is the unit in m/s on Figure 2 for wind velocity, but in Pa/s in Table 3?

Lines 93 – 94: How much data was not used because of the 5 m/s wind threshold. It is my experience that some of the strongest precipitation events occur coincidently with strong winds. You might want to comment how this effects the total scope of precipitation events you are training your model on.

Lines 95 – 102: If you were to extend this work in the future, it might be good to use wet-bulb temperature as a way to split when it is raining vs snowing (Sims and Liu 2015).

Line 152: 90/10 split is sufficient usually, but could you comment on how using a non-shuffled dataset could have seasonality issues? What I mean by that is that often times field campaigns are centered on the event they wish to capture. Thus the bookend times (near the beginning of a campaign and near the end of the campaign), precipitation might be reduced (coming into or out of a 'dry' season). This could be a problem if all of your test splits have weak precipitation events.

Line 276: What p-value and statistical test was used to make the significant conclusion? Please refrain from using the word significant unless you used a statistical test to determine significance.

Lines 306-307: What do you mean by 'assimilate non-attenuated near surface radar data" in the context of spaceborne radars? As you noted before the blind-zone is an issue because of clutter, not attenuation. I am a bit confused by this statement.

Lines 315-316: Just because an echo is > 3 km does not mean it is convective. There are plenty of GPM and CloudSat profiles that have stratiform echoes reaching all the way up to the tropopause (~10 km in the mid-latitudes). Also, the planetary boundary layer in most locations is likely not extending up to 3km. I would guess maybe 1-2 km on average. But I am not an expert in boundary layers. Be careful in the statements here.

Data Availability: Why not cite the NASA data websites here? E.g., https://ghrc.nsstc.nasa.gov/uso/ds_details/collections/gpmgcpxC.html this would be helpful for people to grab the data.

**References:**

Chase, R. J., Harrison, D. R., Burke, A., Lackmann, G. M., & McGovern, A. (2022). A Machine Learning Tutorial for Operational Meteorology, Part I: Traditional Machine Learning, *Weather and Forecasting* (published online ahead of print 2022). Retrieved Jul 18, 2022, from https://journals.ametsoc.org/view/journals/wefo/aop/WAF-D-22-0070.1/WAF-D-22-0070.1.xml

Sims, E. M., & Liu, G. (2015). A Parameterization of the Probability of Snow–Rain Transition, *Journal of Hydrometeorology*, *16*(4), 1466-1477. Retrieved Jul 18, 2022, from https://journals.ametsoc.org/view/journals/hydr/16/4/jhm-d-14-0211_1.xml

---

## Author Comment (AC1)

DeepPrecip: A deep neural network for precipitation retrievals
Atmospheric Measurement Techniques
Aug. 25, 2017

**Reviewer Response Document**

**Reviewer 1**

We thank the reviewer for their detailed examination of the manuscript and especially their focus on improving the clarity of the machine learning components of the article. We have responded to each of their comments below with a description of how we have adjusted the manuscript.

**The main concern I have is with the main result of the paper and the units of the data in Figure 4. To me these accumulation values seem unreasonably small, which makes me concerned there is some sort of error in either a unit conversion or these are truly just really light precipitation events. Let me explain my reasoning. The maximum accumulation reported in Figure 4a (and Figure 6a) shows a 20 min accumulation of 0.1 mm. I am more familiar with English units, so 0.1 mm is 0.004 inches of precipitation. 0.004 inches of precipitation is less than the precision of a common ASOS tipping bucket (1 tip in a tipping bucket gauge is 0.01 inches of precipitation). Thus most of the precipitation events shown in Figure 4 are instances where the amount of precipitation in 20 mins is less than 1 tip of an ASOS rain gauge. I hope you see my concern if these accumulations are correct. If the maximum precipitation amount is less than 1 tip of a tipping bucket gauge how representative is the data the authors present compared to the 'global' distribution of precipitation, especially convective precipitation. While we are discussing Figure 4a, do you have any thoughts of why for JOYCE and Marquette the ML predictions are basically anchored around 0.04 mm? One last comment on Figure 4, (specifically c). I am confused how this plot was made. What is a normalized time step? Could you help readers by explaining a bit more how this plot was made in the text?**

We thank the reviewer for their comment and agree that the presentation of these results and their units were not clearly presented to the reader. The reason values in Fig. 4 typically fell below 0.1 mm is a consequence of the non real time (NRT) accumulation values we use from the Pluvio2 automatic weighing gauge. Unlike the real time (RT) 1-minute intensity measurements from the Pluvio2, the NRT values require a 5 minute delay from the point of observation to provide an accumulation measurement. With this delay, along with improved filtering, finer amounts of precipitation can be captured by the gauge (OTT, 2022). We agree that this information could be communicated more clearly, and we now show the total sum of NRT accumulation over the 20 minute period instead of the average 5 minute NRT values over the 20 minute period. This has the result of increasing the amounts of reported accumulation by a factor of 4, making the results easier to understand and bringing the reported values more closely in line to what is traditionally used in the literature when referring to gauge accumulation quantities. The use of the NRT values has now been explicitly described in the text in Section

4.1, paragraph 1. In terms of Figure 4a, this is an excellent point and we believe this is a consequence of mixed-phase precipitation occurrence at these sites during these periods and an inability of the model to fully capture the resulting precipitation intensity. Both of these periods had temperatures near zero coupled with periods of precipitation extremes and we likely require additional input covariates or more training data for the model to better link the observed atmospheric conditions to a correct intensity. You will note an improved skill in DP over the RF in early JOYCE measurements though, suggesting some improved skill here. We now include an additional description of this feature in the manuscript in Section 4.1, paragraph 1. Regarding your last comment on Fig 4.c, we agree that the "normalized timestep" was not clear, and have now broken this into two plots in a similar manner to Fig 4.a.

**While I am happy the authors made sure they spent the time to explain their hyperparameter search and some details of their training/test splits, there is no explicit comment of which dataset is being shown in the results section. This is vital to any machine learning paper. The authors must state if the results being shown are from the training or the test dataset. This will allow readers to assess if the results are an unbiased assessment of skill or if they seem to be overfit. This gets more challenging since the authors did a k-fold cross validation approach, I am unsure which fold they used to show the results. Please explain.**

The results in this paper are derived from the test set of our 90/10 stratified CV split. We train/test 10 identical DP models based on different non-shuffled splits of the available dataset (always using the test set when referring to model performance). Additionally, each CV split is now run 50 times using dropout to provide additional insight into model uncertainty. To make this clearer to the reader, we have now included an additional paragraph at the end of Section 3.2 where we describe the DeepPrecip testing methodology.

**Upon inspection of the references provided for Table 2, I noticed that some of these are not specifically K-band relationships. For example Kulie and Bennartz (2009) derive relationships for W-, Ka- and Ku- but not K. Similarly, Matrosov (2007) is for Ka and W. Lastly, Marshall and Palmer (1948) is a Rayleigh power law. While it might seem like using a Ka-band relationship for K-band is harmless, issues arise when non-Rayleigh conditions are encountered (e.g., particle sizes are similar to the wavelength), which tend to coincide with large precipitation rates. You should acknowledge that this could be a source of error in your analysis and might be an unfair comparison for your discussion on lines 195 – 209 (Figure 4cd). Lastly, you state in the table caption and the discussion on line 112 that all the relationships are K-band, which is incorrect. To prevent future readers from inaccurately using the reported relationships in your paper on their K-band radars, please correct this mistake.**

We agree that the wording in this section is incorrect and have restructured references to these comparison power laws to reflect the fact that they are derived from similar, but not necessarily exactly the same band as the MRR. We agree that these slight differences in bandwidth are a source of additional uncertainty in our analysis (and now make reference to this with additional

sentences in the final paragraph of Section 5), however based on comparisons from previous studies comparing between similar MRR-derived power laws to KuKa retrievals, we expect errors to be mostly negligible (Kidd et al., 2021, Souverijns et al., 2017; Das & Maitra, 2016; Rakshit & Maitra 2016). We also now explicitly list the derived band for each power law in Table 2.

**The authors mention a random forest model that was based on previous work, but no citation is provided for this model. Given that this random forest model is involved in the primary conclusions of this paper, there is more detail needed. The current description on Lines 146 – 149 is insufficient for reproducibility do not include any of the details of how big the random forest is. Please provide the citation where this model was developed. If there is no citation, please provide more specific details on the random forest model. Also, please note if this model was re-trained on your current data or is it still using the X-band snowfall relationships from GCPEX.**

We thank the reviewer for this comment and have now added a reference to the paper in which the RF model was developed with all training details (https://journals.ametsoc.org/view/journals/apme/aop/JAMC-D-22-0036.1/JAMC-D-22-0036.1.xml) as it is now accepted and published in JAMC. This model was retrained using the same datasets/CV structure as DeepPrecip and this information has been clarified in the manuscript in Section 3.2, paragraph 4.

**There was no discussion on if you scaled the features of the ML model. It is common practice to scale data to have mean 0 and variance 1 in machine learning so that the ML model doesn't unintentionally use a variable with a larger absolute value. For example the dynamic range of radar reflectivity is -10 – 40 dBZ. While the range of temperature is 233 – 313, and the range of doppler velocity is -5 – 5 m/s. See how these three all vary on a different order of magnitude? Did you end up scaling your data? Or did you use batchnorm in training? (I did not see this a parameter in Table 5). Please comment on this.**

We can confirm that we do apply a scaling to have a mean of 0 and variance of 1 for all data before training. This has been clarified in the text in paragraph 5 of Section 3.2.

**It would seem there was an issue with LaTex building the document, all of the parenthetical citations do not correctly put citations into parentheses. This made reading some parts of the paper more difficult. Please be sure to use \citep[e.g.,][]{Paper} to correctly get the formatting to work. (or \citet for inline citations).**

We thank the reviewer for this comment and have updated all references throughout the manuscript to properly use citet and citep commands.

**Line 16: This is an example of the citation issue noted in the minor comments.**

This has now been fixed.

**Lines 60: Be careful here. In my head liquid water content is usually the water content per cubic meter (e.g., g/m^3). Might be good to use a different word here, "records precipitation accumulation" something like that.**

We agree and this has now been changed as suggested by the reviewer.

**Line 84: could you spell out what TMP and WVL are? This is the first time they are defined**

TMP is atmospheric temperature and WVL is vertical wind velocity. The definitions for these variables have now been moved to their first use/definition on this line.

**Figure 2: I assume darker colors mean higher density? You might want to either include a colorbar somewhere or write it in the caption. Could you note in the caption that the wind velocity is vertical wind velocity and which direction negative is? (is negative wind velocities up or down?). This confused me at first because I thought it might be the horizontal wind velocity, but then I didn't know how to interpret negative values. Why is the unit in m/s on Figure 2 for wind velocity, but in Pa/s in Table 3?**

Yes, the reviewer's assumption is correct, the darker colors indicate higher density of observations (we now mention this in the Figure 2 caption). We have also included a more detailed description of wind velocity (i.e. the speed of air motion upwards/downwards using a pressure-based vertical coordinate system) in Section 2.4 of the manuscript. Therefore negative values indicate upwards air motion (since pressure decreases with height). The units have now been updated on Fig. 2 to properly reflect the Pa/s units described in Table 3.

**Lines 93 – 94: How much data was not used because of the 5 m/s wind threshold. It is my experience that some of the strongest precipitation events occur coincidently with strong winds. You might want to comment how this effects the total scope of precipitation events you are training your model on.**

While it varies based on the location, we find that approximately 16% of our available sample is dropped when we apply the 5 m/s wind threshold. We agree that this can lead to a loss of some high intensity precipitation events at certain sites, but examining pre- and post-dropped wind threshold data shows that the maximum intensity precipitation events are not removed after applying this wind thresholding technique. We now make reference to this in Section 2.5, paragraph 1.

**Lines 95 – 102: If you were to extend this work in the future, it might be good to use wet-bulb temperature as a way to split when it is raining vs snowing (Sims and Liu 2015).**

We agree that this would be a good addition in future work and have included this reference in Section 2.5, paragraph 2. In terms of rain snow partitioning, after performing a more extensive model validation using 50 dropout runs per CV split, we re-examined our temperature thresholding and found that the 5 degree C threshold actually provided slightly better overall performance and gave us a larger snowfall sample to work with for comparison purposes later on (note that the main results do not change much). The manuscript has been updated to reflect this change.

**Line 152: 90/10 split is sufficient usually, but could you comment on how using a non-shuffled dataset could have seasonality issues? What I mean by that is that often times field campaigns are centered on the event they wish to capture. Thus the bookend times (near the beginning of a campaign and near the end of the campaign), precipitation might be reduced (coming into or out of a 'dry' season). This could be a problem if all of your test splits have weak precipitation events.**

As a consequence of the extended observational periods from multiple sites (e.g. JOYCE, Ny-Alesund and Marquette all have data over multiple seasons/years), combined with the manner in which we perform the 10-fold CV (training and testing our model on different contiguous sections of the full dataset) stratified by each site, we found that we end up gathering a representative enough sample for the CNN to perform in a robust manner. While the reviewer is correct that an individual split could potentially focus solely on low intensity events as a result of sampling (and therefore perform poorly on high intensity events), when we examine the performance of the model for each of the individual non-shuffled splits in this case, we find similar overall performance for DeepPrecip (suggesting this is not an influential effect).

**Line 276: What p-value and statistical test was used to make the significant conclusion? Please refrain from using the word significant unless you used a statistical test to determine significance.**

We are referring to the Shapley values to describe the importance of input covariates throughout this section and have now updated the language to avoid confusion with statistical significance.

**Lines 306-307: What do you mean by 'assimilate non-attenuated near surface radar data" in the context of spaceborne radars? As you noted before the blind-zone is an issue because of clutter, not attenuation. I am a bit confused by this statement.**

We thank the reviewer for catching this, we are indeed referring to ground clutter here and have now updated the language on this line to reflect that.

**Lines 315-316: Just because an echo is > 3 km does not mean it is convective. There are plenty of GPM and CloudSat profiles that have stratiform echoes reaching all the way up to the tropopause (~10 km in the mid-latitudes). Also, the planetary boundary layer in most locations is likely not extending up to 3km. I would guess maybe 1-2 km on average. But I am not an expert in boundary layers. Be careful in the statements here.**

We agree and have updated this line to simply refer to possible underestimation biases from missing precipitation above the 3 km range of the MRR.

Again, we thank the reviewer for their constructive criticism and for motivating us to further improve the quality and accuracy of the article.

**References:**
Das, S., & Maitra, A. (2016). Vertical profile of rain: Ka band radar observations at tropical locations. Journal of Hydrology, 534, 31–41. https://doi.org/10.1016/j.jhydrol.2015.12.053

Kidd, C., Graham, E., Smyth, T., & Gill, M. (2021). Assessing the Impact of Light/Shallow Precipitation Retrievals from Satellite-Based Observations Using Surface Radar and Micro Rain Radar Observations. Remote Sensing, 13(9), 1708. https://doi.org/10.3390/rs13091708

Rakshit, G., & Maitra, A. (2016). Simultaneous Radar Observations of Vertical Profile of Rain Features from Space and Ground at Ku and Ka Bands at a Tropical Location. MAPAN, 31(4), 291–297. https://doi.org/10.1007/s12647-016-0183-3

Souverijns, N., Gossart, A., Lhermitte, S., Gorodetskaya, I. V., Kneifel, S., Maahn, M., Bliven, F. L., & van Lipzig, N. P. M. (2017). Estimating radar reflectivity—Snowfall rate relationships and their uncertainties over Antarctica by combining disdrometer and radar observations. Atmospheric Research, 196, 211–223. https://doi.org/10.1016/j.atmosres.2017.06.001 "

King, F., Duffy, G., & Fletcher, C. G. (2022). A Centimeter Wavelength Snowfall Retrieval Algorithm Using Machine Learning. Journal of Applied Meteorology and Climatology, 1(aop). https://doi.org/10.1175/JAMC-D-22-0036.1

OTT Hydromet, 2022. Operating Instructions OTT Pluvio2 precipitation gauge, Document number 70.020.000.B.E 04-0515

---

## Author Comment (AC2)

DeepPrecip: A deep neural network for precipitation retrievals
Atmospheric Measurement Techniques
Aug. 25, 2017

**Reviewer Response Document**

**Reviewer 2**

We thank the reviewer for their detailed analysis of our manuscript and the constructive feedback they provided. We have responded to each comment below along with the specific changes we will make to the manuscript.

**Precip phase: Sims and Liu (2015) show a simple scheme for estimating phase – it really depends on wet bulb temperature (Tw), not T, and you've already got the ERA5 data in hand to make the calculation.**

We thank the reviewer for their suggestion and agree that this would be a beneficial technique to incorporate in further iterations of the model with more of a focus on precipitation phase classification. At this point, we feel that incorporating another full column of atmospheric data as an input would likely fall outside of the scope of this current project as it would require an entire new data alignment/preprocessing step, model training and hyperparameterization phase which would be a costly endeavor (in both time and computing resources). However, we have updated the manuscript in Section 2.5, paragraph 2 to include this reference along with a suggestion for further analysis in followup work.

**Undefined magic operations: These need a reference at a bare minimum, preferably with a 1-line "what they do" statement. I can cite: L.161: Adam optimization; L.181: squared errors elbow criterion method; L.214: using dropout; and I would encourage the authors to review the manuscript for other such mystery names.**

We agree that these topics should be described in more detail to make it clear to the reader what is actually being tuned during model development. We now include additional descriptions of each of these phrases along with regularization techniques which were used (i.e. Adam optimization, L2, dropout) where they appear in the manuscript.

**Connect text to the flow chart in Fig. 2b: In particular, it is not clear how Section 3.4 fits into the overall processing workflow.**

We have updated Figure 2.b to highlight where in the model pipeline the clustering occurs, and included a reference to this in the text. We also now include an additional reference to Section 4.2 of the manuscript where the clusters are later used for interpreting feature importances.

**Undercatch: The text keeps alluding to gauge errors, particularly for snow, but never really confronts the beast. Essentially all operational gauges bias low, most acutely for snow. This low bias affects the statistics, and there ought to be at least an organized, if short qualitative statement about this issue.**

We agree that this topic should be discussed in more detail and have now expanded our description of undercatch and the methods we have taken to minimize its impact through additional sentences and references added in Section 2.5, paragraph 1.

**References in the text: These are mostly in a form that is non-standard in my experience (no surrounding parentheses, such as L.16, but not always).**

All references in the text have now been corrected.

**L.50-52: Stating the references here is unnecessary; just point to Table 1.**

We agree and now simply point to Table 1 as suggested by the reviewer.

**L.57: "NRT" is almost always "near-real-time". This is the only place where it's used, so just say "post-real-time" and don't give an acronym.**

We have removed this acronym for clarity.

**L.70-82: The phrasing is awkward; I'd describe the standard situation first and then summarize the deviations.**

We agree that including these results from Ny-Ålesund does not flow well at this point in the manuscript. We have now moved the second paragraph of Section 2.3 to Section 4.2 (to where these details are important for understanding the vertical sub-sectioning analysis we perform).

**Write out all acronyms: TMP, WVL, Ze, S/R, P, RF, MLP, CNN, perhaps plus others.**

We thank the reviewer for this comment and have gone through the manuscript writing out all acronyms upon first use.

**L.84-87: The language can be simplified.**

We agree with the reviewer and this section has now been rephrased and simplified.

**L.122-123: If Cold Lake and Ny-Ålesund are excluded, what *is* done there?**

The data from these sites are still used as part of the training and testing process for DeepPrecip in Section 4.1 and in the uncertainty quantification component of the paper in Section 4.2 when we consider different subsections of the vertical profile and the influence of

these regions on retrieval skill. This additional data helps the model learn about precipitation events that may be unique to certain regional climates. We do not derive Z-P power laws at these sites because of the limited sample at Cold Lake and because of the limited vertical extent at Ny-Ålesund.

**L.183: "event-types" would be more descriptive as something like "intensity classes". And, in subsequent discussions, the same terminology should be used whenever intensity classes are referenced.**

We agree and this terminology has now been adopted here and in the Fig. 3 caption.

**L.203-206: Kindly eliminate the redundancy.**

These lines have now been adjusted for clarity.

**L.214: This is the only mention of mixed phase. If it's going to come up here, it needs to be introduced back when rain and snow events are introduced and related to Fig. 4.**

We agree with the reviewer and now introduce mixed-phase precipitation and the challenges it presents in the first paragraph of Section 4.1, when Figure 4 is introduced.

**L.219-221: This feature isn't really discernable in Fig. 5.**

We agree, and now include additional commentary in the text describing the biases in the Ze-P relationships across multiple sites on these lines.

**Figs. 4a, 5a: I would suggest bolder separation between the different sites so that it's easier to see them as separate time series. Also, the graphic needs to be larger. Finally, the horizontal time axis needs better labeling showing time increments.**

Both Figures 4.a and 5.a have been updated with bolder lines to delineate between different stations. We have also increased the line width and visibility of lines in both figures to make them more readable and added further time information to the x-axis of Fig. 4.a, as requested by the reviewer.

**L.276: I think it's "In all other cases, …"**

This has now been corrected.

**L.280: In addition to SPW, the abbreviations for reflectivity and Doppler velocity should be defined here.**

These definitions have now been included in the text when first defined.

**Fig. 8: In addition to labeling for bin, the Y axis also needs labeling for height, since this is referenced in the text.**

We agree with the reviewer and have added an additional y-axis for height to Fig. 8.

**L.301: "assumption" isn't quite the right word; "speculation"?; "prior inference"?**

We have updated this language to "prior inference" as suggested by the reviewer.

**L.305-308: This last sentence needs to be more straightforward. I think you're suggesting more aggressive use of the un-blanked, or at least, less generously blanked, satellite radar data.**

This sentence has now been updated for clarity as suggested by the reviewer.

**L.323-324: The name usually includes "mission" as "Global Precipitation Measurement (GPM) mission".**

The word "mission " has now been added on this line.

We thank the reviewer once again for their constructive comments and suggested improvements for further enhancing the quality and clarity of the article.

---

## Referee Report (RR1)

Title: DeepPrecip: A deep neural network for precipitation retrievals

Correspondence: Fraser King (fdmking@uwaterloo.ca)

This paper presents a deep-learning model for surface precipitation accumulation retrieval using near-surface vertical column radar reflectivities and environmental parameters from ERA-5. One of the major contributions of this work is its improved high accuracy compared to traditional pow-law relationships and a less complex RF model. Another major contribution is that the study not only analyzed the vertical column structure up to 3 km above the surface as a whole to retrieve precipitation accumulation, but also investigated the lower and upper vertical layers of profiles. The quantitative analysis of the comparison then help them to conclude that the combination of both layers can achieve the maximum retrieval accuracy. These make this paper matches the aim and scope of ATM.

In view of the other two reviewers' comments, the revised manuscript has been largely improved in a strict and thorough way. The topic of this paper is very interesting and important for continued investigation of using emerging techniques to improve remote sensing precipitation estimation. With sufficient literature review, the authors first acknowledge the current available methods as well as their advantages and drawbacks, and then identify the research gap and propose their new method for filling the gap. The data selection and methodology are clearly described. Detailed and quantitative result analysis provide enough supporting evidence for their reasoning. As a result, I recommend this manuscript to be accepted with minor revision.

Minor comments:

Line 4: it should be "develop a … retrieval algorithm …".

Line 108: references format needs correction.

Line 135: please write the full name of RF as this is the first times it is used rather than in Line 203.

Figure 4: I understand there is a standard deviation bar shown in (b), but it seems like it also appears in (a) since the authors state "1 standard deviation … shown in the shaded regions"? If this is true, I don't see the standard deviation bar or shaded region in (a) … Or is it actually referring to (c) and (d) rather than (a) where there are multiple light red and blue lines not explained?

---

## Author Response (AR2)

DeepPrecip: A deep neural network for precipitation retrievals
Atmospheric Measurement Techniques
Sept. 26, 2022

**Editor Response Document**

We thank the editor for their assessment of our work and their additional constructive comments for improving the manuscript quality. We have provided a point-by-point response below.

**Line 4: it should be "develop a ... retrieval algorithm ...".**

We agree, and this has now been added on line 4.

**Line 108: references format needs correction.**

These references have now been corrected on line 108.

**Line 135: please write the full name of RF as this is the first time it is used rather than in Line 203.**

We have now shifted the definition of RF (random forest) to line 135 as requested.

**Figure 4: I understand there is a standard deviation bar shown in (b), but it seems like it also appears in (a) since the authors state "1 standard deviation ... shown in the shaded regions"? If this is true, I don't see the standard deviation bar or shaded region in (a) ... Or is it actually referring to (c) and (d) rather than (a) where there are multiple light red and blue lines not explained?**

We thank the editor for their comment and agree that this should be described better in the figure caption. The shaded region we are referring to here is present in (a), it is just fairly subtle. The presence of the region is most notable at the start of JOYCE for instance, and we have highlighted this in the text to help guide the reader. We have also included additional text in the caption describing what the red and blue lines refer to in (c) and (d).